computational chemistry/chemical biology/ bioinformatics

antimicrobial resistance, *Acinetobacter baumannii*, phytochemicals, molecular docking

**Author for correspondence:**
Shahab Shahryari
e-mail: sha.shahriari@yahoo.de

This article has been edited by the Royal Society of Chemistry, including the commissioning, peer review process and editorial aspects up to the point of acceptance.

# Screening of anti-*Acinetobacter baumannii* phytochemicals, based on the potential inhibitory effect on OmpA and OmpW functions

Shahab Shahryari[1], Parvin Mohammadnejad[2] and Kambiz Akbari Noghabi[1]

[1]Department of Energy and Environmental Biotechnology, National Institute of Genetic Engineering and Biotechnology (NIGEB), PO Box 14155-6343, Tehran, Iran
[2]Division of Agricultural Biotechnology, National Institute of Genetic Engineering and Biotechnology (NIGEB), PO Box 14965/161, Tehran, Iran

SS, 0000-0003-3484-3500; PM, 0000-0002-7250-2910; KAN, 0000-0002-6745-2795

Therapeutic options including last-line or combined antibiotic therapies for multi-drug-resistant strains of *Acinetobacter baumannii* are ineffective. The outer membrane protein A (OmpA) and outer membrane protein W (OmpW) are two porins known for their different cellular functions. Identification of natural compounds with the potentials to block these putative porins can attenuate the growth of the bacteria and control the relating diseases. The current work aimed to screen a library of 384 phytochemicals according to their potentials to be used as a drug, and potentials to inhibit the function of OmpA and OmpW in *A. baumannii*. The phytocompounds were initially screened based on their physico-chemical, absorption, distribution, metabolism, excretion and toxicity (ADMET) drug-like properties. Afterwards, the selected ligands were subjected to standard docking calculations against the predicted three-dimensional structure of OmpA and OmpW in *A. baumannii*. We identified three phytochemicals (isosakuranetin, aloe-emodin and pinocembrin) possessing appreciable binding affinity towards the selected binding pocket of OmpA and OmpW. Molecular dynamics simulation analysis confirmed the stability of the complexes. Among them, isosakuranetin was suggested as the best phytocompound for further *in vitro* and *in vivo* study.

# 1. Introduction

In recent years, antimicrobial resistance (AMR) by Gram-negative bacteria has become a serious problem with a possible devastating impact on the economy and human life [1]. With the advent of the post-antibiotic era, the need to use more effective and safer antimicrobial compounds is fully felt. [2]. As one of the six superbug ESKAPE pathogens, *Acinetobacter baumannii* is a naturally transformable Gram-negative pathogen with increasing clinical importance [3]. Multi-drug-resistant (MDR) strains of *Acinetobacter* are responsible for chronic infections and can rapidly confer resistance to most of the currently used antibiotics. Thus, *A. baumannii* is in the global priority pathogen list to develop effective antimicrobial therapies against it [4]. Due to the rapid changes in the genetic constitution of *A. baumannii* and lack of enough knowledge about resistance mechanisms, no efficient antibiotic against this microorganism has yet been developed by the pharmaceutical industry [5]. This paucity of effective antibiotics has revived scientific interest in finding efficient antimicrobial agents, with the ability to kill, inhibit the growth or inhibit the activity of some essential virulence factors of *A. baumannii* [6–9].

The outer membrane proteins (OMPs) of Gram-negative bacteria have attracted some attention as drug targets due to their extensive functions [10,11]. Outer membrane protein A (OmpA) as the most abundant slow porin in members of genus *Acinetobacter* plays different rules in cells [12,13]. This protein is closely associated with the virulence and survival of the bacteria under harsh conditions [14–16]. OmpA structure is composed of a β-barrel and a periplasmic (OmpA-like) domain. In addition to a structural role for OmpA of *A. baumannii*, it can play an important role in biofilm formation and adhesion to host cells [12,17–21]. The absence of OmpA in *A. baumannii* and *Escherichia coli* is associated with significant reduction of virulence [22,23]. Also, it has been suggested that four external loops, connecting N-terminal β-strands, participate in some interactions with host cells (or other surfaces), and can be used as drug targets [24].

Outer membrane protein W (OmpW) is another porin with a pivotal role in the uptake of nutritional substances such as iron [25]. The role of iron acquisition in the pathogenicity of *A. baumannii* has been reported previously [26], and disruption of this system can be effective in treating infections caused by this microorganism [27–29]. Decreased minimum inhibitory concentrations of several antibiotics due to deletion of OmpW from *E. coli* [30], as well as reduced intestinal colonization of Δ*ompW* mutant strains of *Vibrio cholerae* [31], are among the indications that OmpW can be served as a potential drug target in Gram-negative bacteria. *In silico* and *in vitro* studies on OmpW resulted in the identification of three inhibitors of this porin in *A. baumannii* [32]. Therefore, any molecule with the ability to disrupt the functions of OmpA and OmpW can help treat the disease caused by this bacterium.

Researchers have reported the benefits of infectious disease treatment using poly-pharmacological drugs [33]. In the light of this, the aim of the current study was *in silico* screening of the plant-based compounds with potential anti-*A. baumannii* activities. The adhesion property of OmpA and the function of OmpW in the uptake of nutrients were targeted for the screening of compounds. For this purpose, a library consisting of 384 plant-based compounds, known as an anti-bacterial or anti-viral compound, was selected for this study [9,34–36]. We applied *in silico* approaches to prioritize these biomolecules for toxicological evaluation and safety concerns [37,38]. The phytocompounds with the highest affinity for both OmpA and OmpW were selected as the multi-target drug candidates. The results for molecular docking were further evaluated using molecular dynamics (MD) simulation. Our results led to the identification of three potential mutual OmpA and OmpW functional inhibitors, isosakuranetin, aloe-emodin and pinocembrin. Isosakuranetin was suggested for further *in vitro* and *in vivo* studies.

# 2. Data and methodology

## 2.1. Tools and servers

A list of all tools and servers used in this work with a brief description is illustrated in electronic supplementary material, table S1.

## 2.2. Targets sequences

The complete amino acid sequences of OmpA (BAN86529 accession no.) and OmpW (BAN89067 accession no.) were retrieved from the GenBank database at National Center for Biotechnology Information (NCBI) (http://www.ncbi.nlm.nih.gov/protein/) in FASTA format.

## 2.3. Signal peptide prediction

The secretary nature of selected proteins was predicted using SignalP 4.1 server (http://www.cbs.dtu.dk/services/SignalP/), which integrates a prediction of cleavage sites and signal or non-signal peptide prediction based on a combination of several artificial neural networks. LipoP 1.0 server (http://www.cbs.dtu.dk/services/LipoP) was used to predict probable signal peptide within the sequence and signal peptidase I or II cleavage sites within the protein.

## 2.4. Homology modelling, model refinement and energy minimization

As the full three-dimensional structure of none of the proteins was available in the Protein Data Bank (PDB), a homology modelling was performed to predict the three-dimensional structure of each protein using RaptorX web server (http://raptorx.uchicago.edu/StructurePrediction/predict/). The server predicts the absolute global quality and comparable global quality for each of the residues of the query sequence. In order to check the three-dimensional structure, we used PyMOL molecular graphics system v. 1.7.4.4 (Schrödinger, LLC, Portland, OR, USA). To perform structure refinement and energy minimization of the three-dimensional modelled protein structures, the online server GalaxyRefine (http://galaxy.seoklab.org/) and YASARA software were used, respectively. GalaxyRefine employs the CASP10 assessment to refine the query structure, improving the structural and global quality of the three-dimensional model. This method initially rebuilds side chains and performs side-chain repacking and subsequently uses MD simulation to achieve overall structure relaxation.

## 2.5. Validation of the three-dimensional structures

To validate the refined and optimized three-dimensional structures, three freely available web tools of RAMPAGE (http://mordred.bioc.cam.ac.uk/rapper/~rampage.php), ProSA-web (https://prosa.services.came.sbg.ac.at/prosa.php) and ERRAT (https://servicesn.mbi.ucla.edu/ERRAT/) were used. The server RAMPAGE evaluates the Ramachandran plot by applying PROCHECK principles. The ProSA validation method evaluates model accuracy and statistical significance with a knowledge-based potential. It plots overall excellence scores of the errors calculated in the query 3D structure. Further, the ERRAT server was used to check the reliability of modelled proteins and their overall quality factors.

## 2.6. Prediction of potential ligand-binding pocket

To avoid blind docking and increase the accuracy of molecular docking, the top three possible binding sites of the proteins were identified using METAPOCKET v. 2.0 (https://projects.biotec.tu-dresden.de/metapocket/index.php). As a META server, METAPOCKET is a popular consensus method, which combines eight methods of asLIGSITEcs, PASS, QSiteFinder, SURFNET, Fpocket, GHECOM, Concavity and POCASA to predict binding sites based on three-dimensional protein structures.

## 2.7. Natural compound ligand library

A library of 384 phytochemicals (electronic supplementary material, table S2) with potential anti-infection properties was used in this study [9,34–36,39]. The three-dimensional structures of the molecules were retrieved from UniProt database. In the case of lacking X-ray crystallography or NMR structure of the compound, a three-dimensional conformer was obtained from PubChem database. The ligands three-dimensional structures were energy minimized using Chem3D software before docking. The ligand files were finally saved in PDB format for molecular docking studies.

## 2.8. Pre-filtration and pharmacokinetic analysis of phytochemicals

A compound must pass through multiple filters to be considered as a novel drug. All of the bioactive candidates used in this study were evaluated based on their important physico-chemical properties using SwissADME (http://www.swissadme.ch/) online server. Initially, those compounds satisfying Lipinski's rule of five (RO5) were selected and those that violated the rule were eliminated from downstream analysis. Pharmacokinetic (PK) properties of absorption, distribution, metabolism, excretion and toxicity (ADMET) with a crucial role in the development of drug design were predicted using the pkCSM webserver at (http://biosig.unimelb.edu.au/pkcsm/) to decrease the failure rate of the

compound for further analysis. For further screening, PreADME (https://preadmet.bmdrc.kr/) server was used to filter the compounds based on their drug-likeliness, ADME properties and toxicity.

## 2.9. Molecular docking analysis

Molecular docking analysis was performed using AutoDock Vina software [40]. The .pdbqt file of ligands was prepared after specifying of rotatable, non-rotatable bonds and torsion angles in the compounds using AutoDock tools. Grid boxes were defined according to the selected binding pockets and the aim of this study. The protein (receptor) files in .pdbqt format were prepared for a flexible body docking using AutoDock Tools software. As the target for OmpA was external loops because of their potential for adhesion, all the residues of external loops in the predicted binding pocket were chosen for torsions. Concerning OmpW, only the residues of the binding pocket located towards the β-barrel were selected for torsion, because of the aim of this study for screening of compounds with the ability to block the OmpW channel. The docking process was performed after preparing a configuration file. The obtained interactions and generated poses were visualized using PyMOL. The docking results were evaluated for their orientation, number of hydrogen bonds, number of interacting residues, binding affinity (kcal mol$^{-1}$) and RSMD.

## 2.10. Drug–target interactions and mechanism of binding

The mechanism of binding in drug–target complexes was profiled using the LigPlot$^+$ program (https://www.ebi.ac.uk/thornton-srv/software/LigPlus/). Hydrogen bonds as well as hydrophobic interactions between potential drug molecules and the protein target were studied via two-dimensional interaction diagrams of the docked drug–target complexes.

## 2.11. BOILED-Egg model for prediction of gastrointestinal absorption and blood–brain barrier penetration

To reveal the capability of gastrointestinal absorption (GI) absorption and permeability of the blood–brain barrier (BBB), the BOILED-Egg model of the potential drugs was predicted using SwissADME (http://www.swissadme.ch/).

## 2.12. Molecular dynamics simulation

MD simulation was used to refine the four complexes obtained from molecular docking simulation. MD analysis was performed using a GROMACS 4.6.5 with GROMOS96 force field and simple point charge as a water model [41].

To mimic the experimental conditions, the simulation was carried out at 343 K. Some molecules of water were randomly replaced by ions for neutralization. All systems were simulated under the NpT ensemble within periodic boundary conditions. The neighbour list was updated with a 10-step frequency using the grid search method. The leap-frog algorithm was used with a 2 fs time step. Protein and solvent were coupled separately to a heat bath at the desired temperature with time constant $\tau$T = 0.1 ps applying V-rescale thermostat [42]. After being neutralized, the system was submitted to energy minimization applying the steepest descent algorithm. After that, the equilibrations of systems were done under NVT up to 100 ps at 300 K with restraint forces of 1000 kJ mol$^{-1}$, followed by 100 ps under NPT at the pressure of 1 bar and with restraint forces of 1000 kJ mol$^{-1}$ using modified Berendsen thermostat and Parinello-Rahman barostat algorithms, respectively. The electrostatic interactions were analysed using the particle mesh Ewald method. Ultimately, the MD run was performed with no restraint for 50 ns.

# 3. Results and discussion

## 3.1. Homology modelling, model refinement and energy minimization

Multiple templates were used to predict the three-dimensional structure of OmpA and OmpW (figures 1 and 2). Crystal structure of OmpA-like domain from *A. baumannii* (3TD3) with *p*-value $6.35 \times 10^{-6}$ was used as a template for prediction of OmpA periplasmic domain (figure 1). Five β-barrel crystal structures (1P4T, 2X4M, 3NB3, 3QRA and 4RLC) were suggested for three-dimensional structure prediction of this domain in *A. baumannii* OmpA (figure 1). The crystal structure 4RLC with *p*-value $2.05 \times 10^{-14}$ was used as the best template. The tertiary structure of OmpW predicted to contain only

**Figure 1.** Cartoon representation of the best structures of OmpA modelled by RaptorX and refined by GalaxyRefine (*a*). Amino acid sequence alignment of OmpA and its three-dimensional structure template for the periplasmic domain (*b*). Multiple sequence alignment of OmpA and its three-dimensional structure templates for the β-barrel structure (*c*).

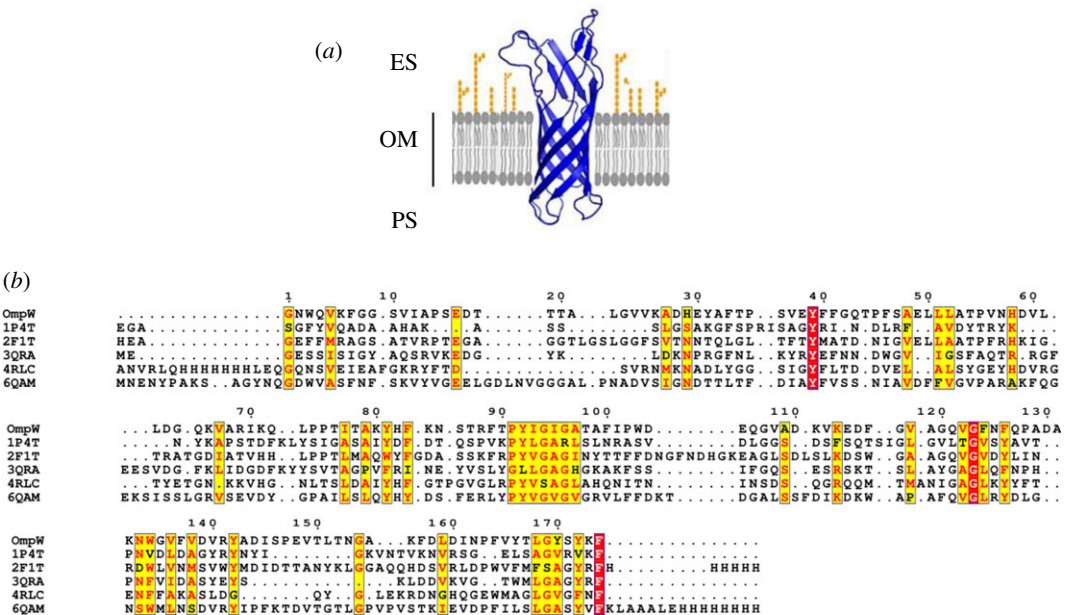

**Figure 2.** Cartoon representation of the best structures of OmpW modelled by RaptorX and refined by GalaxyRefine (*a*). Multiple sequence alignment of OmpW and its three-dimensional structure templates (*b*).

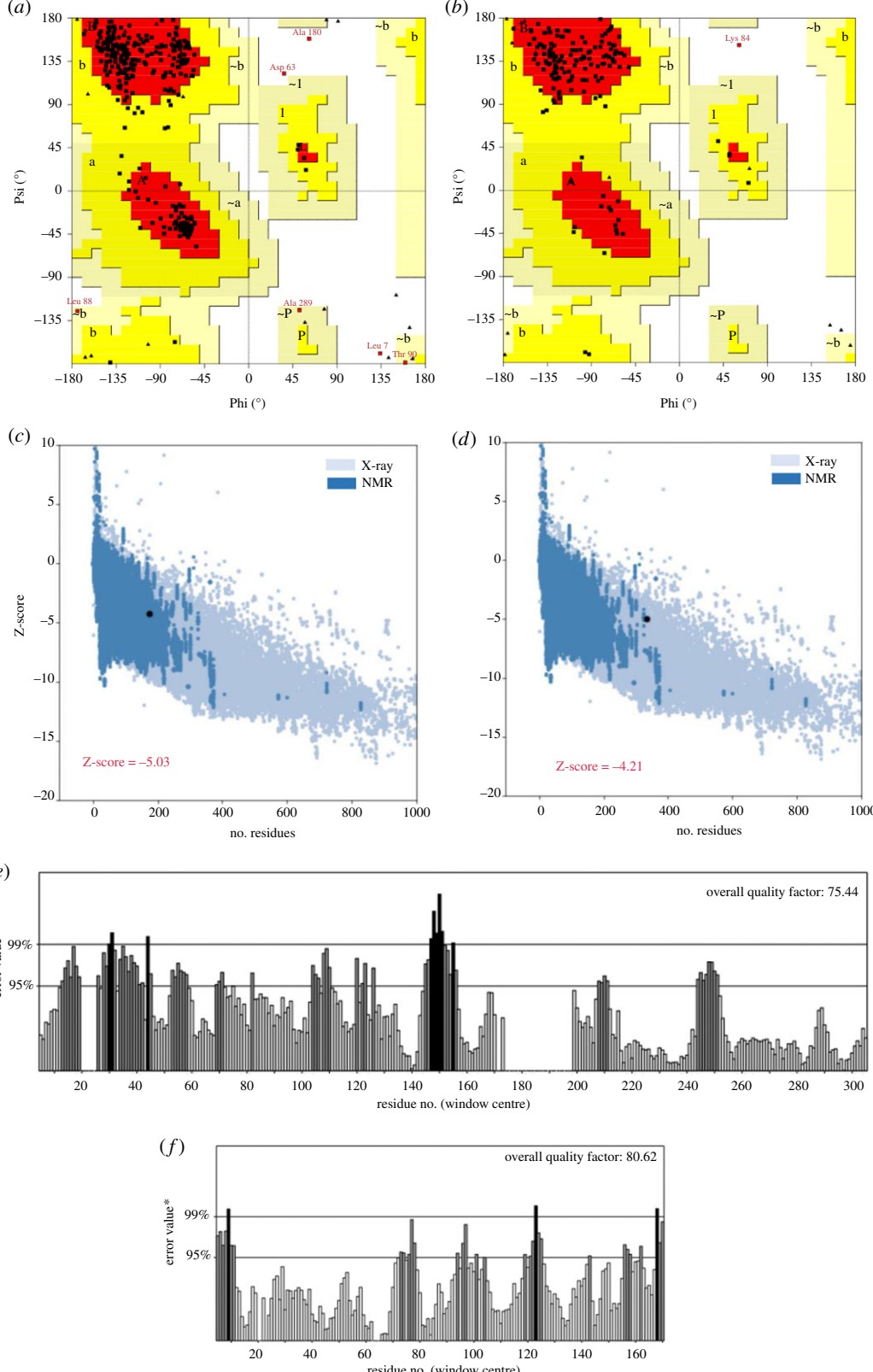

**Figure 3.** Ramachandran plot, the PROSA-web and the ERRAT server validation of the modelled three-dimensional structures of OmpA and OmpW. Distribution of each amino acid in the favoured, allowed and disallowed regions are shown in corresponding Ramachandran plots of OmpA (*a*) and OmpW (*b*). Ramachandran plot summary shows 96.7% of residues of OmpA are in most favourable regions, 1.5% in allowed regions and 1.8% in disallowed regions. For OmpW, 97.1% of residues are in most favourable regions, 2.3% in allowed regions and 0.6% in disallowed regions. Validation of OmpA (*c*) and OmpW (*d*) modelled structures using PROSA-web shows Z-score values as −5.03 and −4.21 for modelled proteins. The overall quality factors obtained by ERRAT for OmpA (*e*) and OmpW (*f*) were 75.44% and 80.62%, respectively.

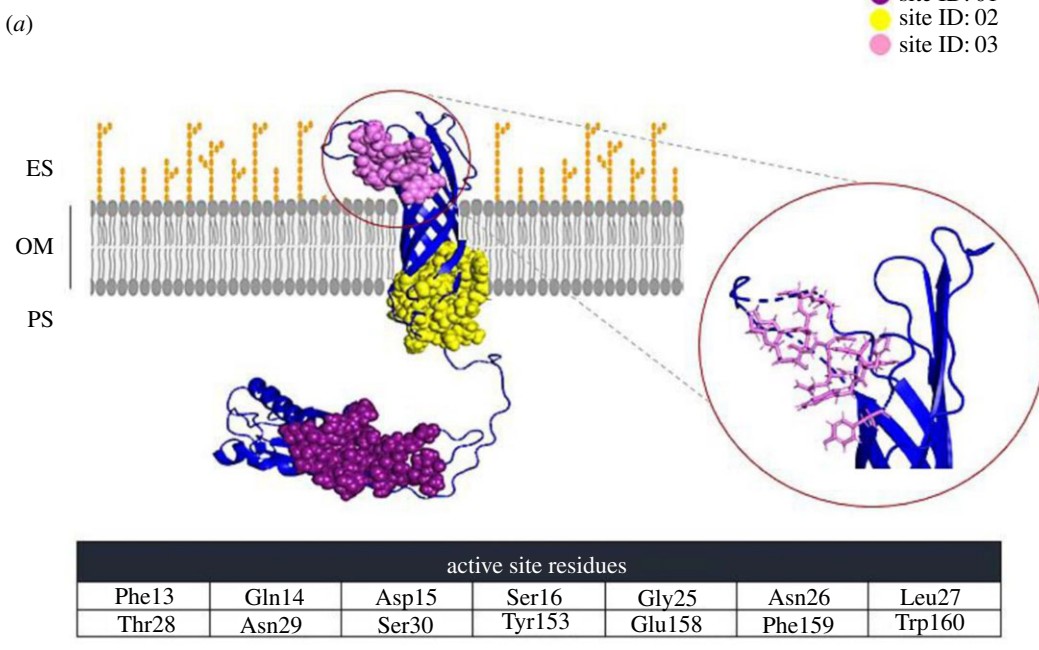

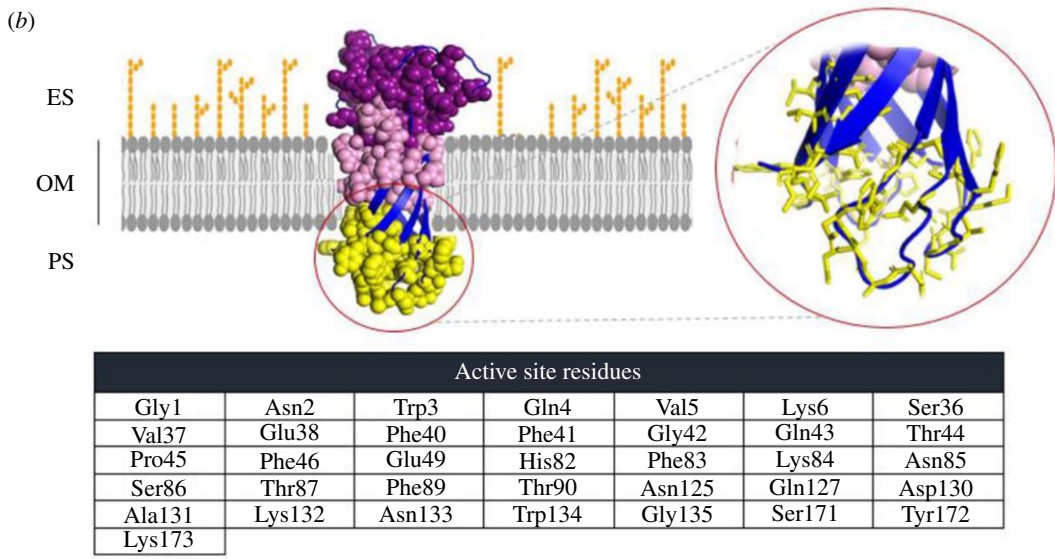

site ID: 01
site ID: 02
site ID: 03

**active site residues (a)**

| Phe13 | Gln14 | Asp15 | Ser16 | Gly25 | Asn26 | Leu27 |
|-------|-------|-------|-------|-------|-------|-------|
| Thr28 | Asn29 | Ser30 | Tyr153 | Glu158 | Phe159 | Trp160 |

**Active site residues (b)**

| Gly1 | Asn2 | Trp3 | Gln4 | Val5 | Lys6 | Ser36 |
|------|------|------|------|------|------|-------|
| Val37 | Glu38 | Phe40 | Phe41 | Gly42 | Gln43 | Thr44 |
| Pro45 | Phe46 | Glu49 | His82 | Phe83 | Lys84 | Asn85 |
| Ser86 | Thr87 | Phe89 | Thr90 | Asn125 | Gln127 | Asp130 |
| Ala131 | Lys132 | Asn133 | Trp134 | Gly135 | Ser171 | Tyr172 |
| Lys173 | | | | | | |

**Figure 4.** Predicted top three potential binding pockets with corresponding active site residues for OmpA (*a*) and OmpW (*b*) by METAPOCKET 2.0. The selected potential binding pockets are emphasized by circle and their residues are shown in table.

one domain (β-barrel domain) and crystal structure 2F1T with *p*-value $3.48 \times 10^{-07}$ was employed as the best template for this prediction (figure 2). Refinement of the obtained 'crude' models from the GalaxyRefine server resulted in five models for each protein (electronic supplementary material, table S3). Based on model quality scores for all refined models, model two with GDT-HA value of 0.9633, RMSD of 0.384, and MolProbity of 1.922, clash score of 17.7 and poor rotamers score of 0.7 was found to be the best model of OmpA. Its Ramachandran plot score was 97.0%. For OmpW, model three was found to be the best model based on GDT-HA of 0.9684, RMSD of 0.392, MolProbity of 1.746, clash score of 11.8 and poor rotamers of 0.0. Ramachandran plot score was 97.1% for this model. YASARA energy minimization tool was used to further improve the quality of the models.

## 3.2. Tertiary structure validation

The validity of the refined three-dimensional modelled structures of OmpA and OmpW was checked by using RAMPAGE, ProSA-web and ERRAT servers (figure 3). The Ramachandran plot checked the

**Table 1.** Molecular docking analysis of the top three compounds which have highest affinity for OmpA. The asterisks represent the compounds with the potential to inhibit OmpA according to previous studies.

| compounds | binding energy (kcal mol$^{-1}$) | RMSD | number of H-bounds | number of interacting residues | interacting residues |
|---|---|---|---|---|---|
| *AOA-2 | −8.5 | 0 | 4 | 12 | Asp15, Lys23, Gly25, Asn26, Leu27, Thr28, Asn29, Ser30, Pro31, Glu157, Glu158, Phe159 |
| *epiestriol | −6.6 | 0 | 1 | 6 | Gly40, Tyr57, Asn58, Gln59, Gln75, Lys76 |
| isosakuranetin | −7.8 | 0 | 1 | 7 | Gln14, Asp15, Leu27, Thr28, Ser30, Glu158, Trp160 |
| aloe-emodin | −7.7 | 0 | 2 | 6 | Gln14, Asp15, Leu27, Thr28, Glu158, Trp160 |
| pinocembrin | −7.5 | 0 | 1 | 7 | Gln14, Asp15, Leu27, Thr28, Ser30, Glu158, Trp160 |

stereochemical quality of the refined proteins. Analysis of the plot for the modelled OmpA revealed that 321 residues (96.7%) of the protein are in favoured regions, which confirms that the model is characterized by stereochemical parameters of a stable structure. This is consistent with the 97.0% score predicted by the GalaxyRefine analysis. Additionally, five residues (1.5%) were predicted to be in allowed regions and six residues (1.8%) in the disallowed region. The same analysis for OmpW verified that 167 residues (97.1%) are in the favoured region, four residues (2.3%) are in allowed and only one residue (0.6%) is in the outlier area. The obtained result for residues of the favoured region is the same as the score obtained by the GalaxyRefine analysis. These results indicate the correct geometry and three-dimensional arrangement of the models. The server ProSA-web further evaluated the overall quality of the models by providing a Z-score, as an indication of overall model quality [43]. The reference ranges for this score change depending on the size of the protein, where the more negative the Z-score, the better is the protein model. The values for OmpA and OmpW were −5.03 and −4.21, respectively. The Z-score of both input structures was within the range of the scores typically found for native proteins of similar sizes, which shows the modelled proteins fall within the range of X-ray solved protein structures. Further verification with PROCHECK showed a good resolution (normality) of 1.5 Å for the structures of both proteins, since most high-resolution X-ray structures have a resolution within 1.5 and 2.0 Å. The ERRAT server was used to check the statistics of non-bonded interactions between different atom types, the reliability of model proteins and the overall quality factor; where higher scores indicate higher quality of protein structure (acceptable range = score greater than 50). The overall quality factors were 75.44% for OmpA and 80.62% OmpW.

## 3.3. Pre-filtration and pharmacokinetic analysis of phytochemicals

Due to the large attrition rate of the molecules under clinical trials, and to ensure that selected compounds can be used as a drug, the putative antimicrobial phytocompounds were screened using PreADMET, SwissADME and pkCSM servers. Different filters, including Lipinski's rule (molecular weight ≤ 500 Da, log p (lipophilicity log) ≤ 5, number of hydrogen bond donor (HBD) ≤ 5 and hydrogen bond acceptor (HBA) ≤ 10, number of rotatable bonds (nRot) ≤ 10, topological polar surface area (TPSA) ≤ 150 Å$^2$) [44], CMC-like rule, MDDR-like rule, lead-like rule and WDI-like rule, were used to select compounds. Moreover, one important parameter for oral bioavailability is molar refractivity (MR) described by the Ghose filter [45]. The Ghose filter quantitatively characterizes small molecules based on computed physico-chemical property profiles that include log P, MR, molecular weight and the number of atoms. According to this rule, the value of MR should lie between 40 and 130 for drug-likeness.

Effective and safe drugs exhibit a good combination of pharmacodynamics and PK including adequate absorption, distribution, metabolism, excretion and tolerable toxicity (ADMET), which reduces the number

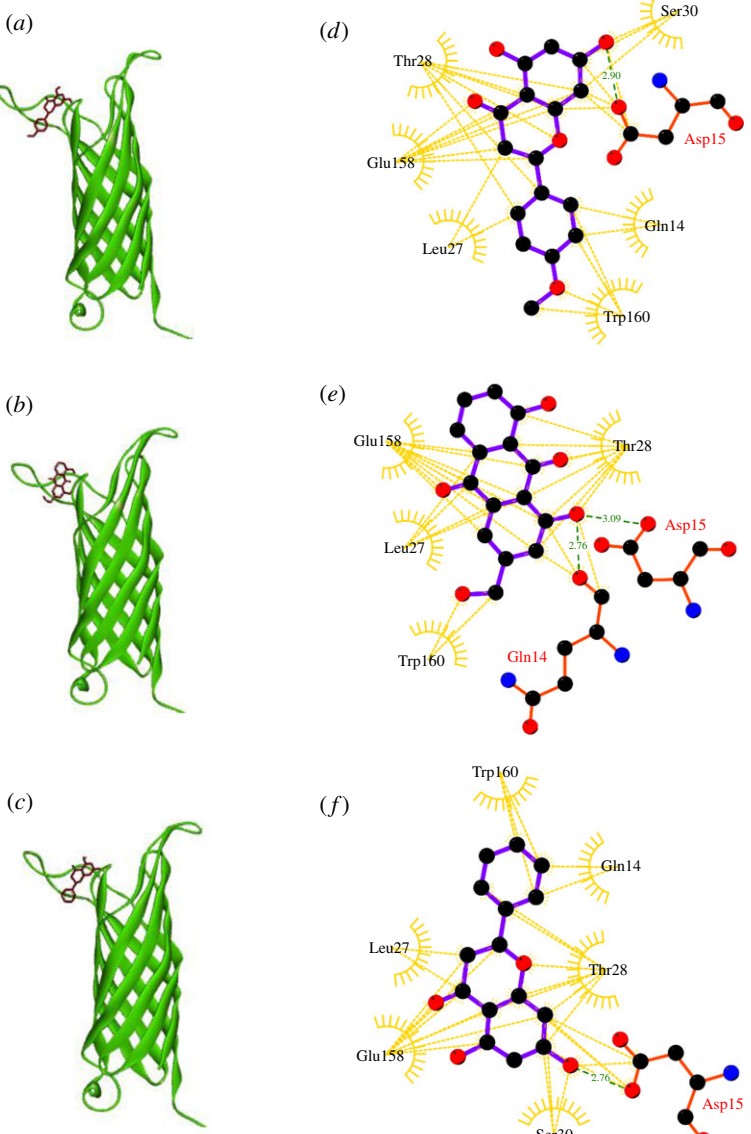

**Figure 5.** Three-dimensional representation of the OmpA-ligand complexes. (a) represents isosakuranetin and OmpA complex, (b) aloe-emodin and OmpA complex and (c) pinocembrin and OmpA complex. (d), (e) and (f) represent two-dimensional interaction diagram of isosakuranetin, aloe-emodin and pinocembrin in the binding pockets of the target, respectively. Ligands, H-bond interacting residues and hydrophobic interacting residues are shown in purple, red and golden, respectively. Green bonds represent hydrogen bonds and golden bonds represent hydrophobic interactions.

of synthesis-evaluation cycles and more expensive late-stage failures by excluding inappropriate drug candidates [46,47]. Regarding this, a compound should be absorbed easily through the gastrointestinal tract so that it can be available in the systemic circulation [48]. To predict the absorption of orally administered drugs, permeability coefficient across monolayers of the human colon carcinoma cell line Caco-2 is commonly used. This permeability as log Papp ($10^{-6}$ cm s$^{-1}$) rate is considered high if log Papp is greater than 0.9 and low if less than 0.9. For the achievement of an optimal clinical drug, a molecule should have high log Papp. The BBB and central nervous system (CNS) permeability values were also predicted for the candidates. It has been reported that those with log BB value greater than 0.3 have the potential to pass the BBB, while those with values less than −1 are poorly distributed to the brain. In addition, candidates that have log PS value greater than −2 are considered to penetrate the CNS, while those with values less than −3 have difficulty in penetrating the CNS.

Considering the binding status of a drug with the blood proteins, the efficiency of a drug is affected by the binding efficacy of the molecule with whole blood proteins [48]. In this regard, we evaluated the fraction unbound (Fu) and steady-state volume of distribution (VDss) for the candidates. In distribution properties, VDss represents the total volume required by the drug concentration to be uniformly

**Table 2.** The ligand-OmpW molecular docking analysis of selected compounds. The asterisks represent the compounds with the potential to inhibit OmpW according to previous studies.

| compounds | binding energy (kcal mol$^{-1}$) | RMSD | number of H-bounds | number of interacting residues | interacting residues |
|---|---|---|---|---|---|
| *D6 | −8.4 | 0 | 1 | 12 | Asn2, Gln4, Phe40, Gly42, Gln43, Thr44, His82, Lys84, Asn85, Thr90, Asn125, Gln127 |
| *epiestriol | −6.7 | 0 | 0 | 6 | Gly1, Trp3, Tyr39, Phe40, Phe41, Ala48 |
| isosakuranetin | −9.8 | 0 | 4 | 14 | Asn2, Gln4, Phe40, Gly42, Gln43, Thr44, His82, Lys84, Asn85, Thr90, Gln127, Asn133, Trp134, Ser171 |
| aloe-emodin | −9.3 | 0 | 3 | 10 | Asn2, Gln4, Phe40, Gln43, Thr44, Lys84, Asn85, Thr90, Gln127, Trp134 |
| pinocembrin | −10.2 | 0 | 1 | 11 | Asn2, Gln4, Phe40, Gly42, Gln43, Asn85, Asn125, Gln127, Asn133, Trp134, Ser171 |

distributed, to give the same concentration at the target site as in the plasma. Higher values of VDss indicate that most of the drug is contained in tissues rather than in plasma (VDss (human) ($\log_{L/kg}$); low if $\log_{L/kg} < -0.15$ and high if $\log_{L/kg} > 0.45$).

Early *in silico* prediction of toxicity endpoints of the candidates was performed. These endpoints consist of the inhibition of cytochrome P450 (CYPs) monooxygenase enzymes [49]. Metabolism enzymes should metabolize a drug candidate to prevent undesirable adverse effects. In this regard, the inhibition of the major human cytochrome P450 isoform of CYP2D6, which is involved in drug metabolism, as well as the renal OCT2 substrate was assessed categorically.

The factors of AMES toxicity and $LD_{50}$ (lethal rat acute toxicity) value of the phytocompounds were investigated using the pkCSM *in silico* screening approach. Among major toxicity endpoints, $LD_{50}$ is an index determination of medicine and poison's virulence. Hepatotoxicity, skin sensitization, and inhibition of hERG potassium ion channel effects were also determined for the evaluation of drug–drug interactions. Moreover, the solubility of the compounds was evaluated by topological methods of ESOL, Ali and fragmental method of SILICOST-IT [48,50].

Results showed that out of 384 investigated compounds, only 27 compounds satisfy the key parameters of these physico-chemical properties (electronic supplementary material, tables S4 and S5). They fulfilled Lipinski's rule of five, CMC-like rule, lead-like rule and WDI-like rule, with no violation, which are beneficial to assess *in vivo* abilities of any compounds. The ADMET profile of these candidates showed that all of the screened compounds possess high intestinal absorption. Near 63% of them showed high log Papp values. Moreover, no hepatotoxicity characterized by disrupted normal liver functions, skin sensitization, hERGI inhibition was observed for them. Our results demonstrated an acceptable prediction of physico-chemical properties in comparison to Lipinski's RO5. It was observed that selected molecules fall within most of the accepted range of the parameters of ADMET. Therefore, they were further screened against target proteins (OmpA and OmpW).

## 3.4. Protein–ligand interactions and mechanisms of binding

Previous studies have suggested that a cyclic hexapeptide (AOA-2) and dichlorophen (D6) can fight against *A. baumannii* by affecting the activity of OmpA and OmpW, respectively [24,32,51]. Therefore, we used these compounds as controls to evaluate the results of molecular docking. AOA-2 prevents invasion of host cells by potential binding to external loops of OmpA β-barrel structure. D6 can locate

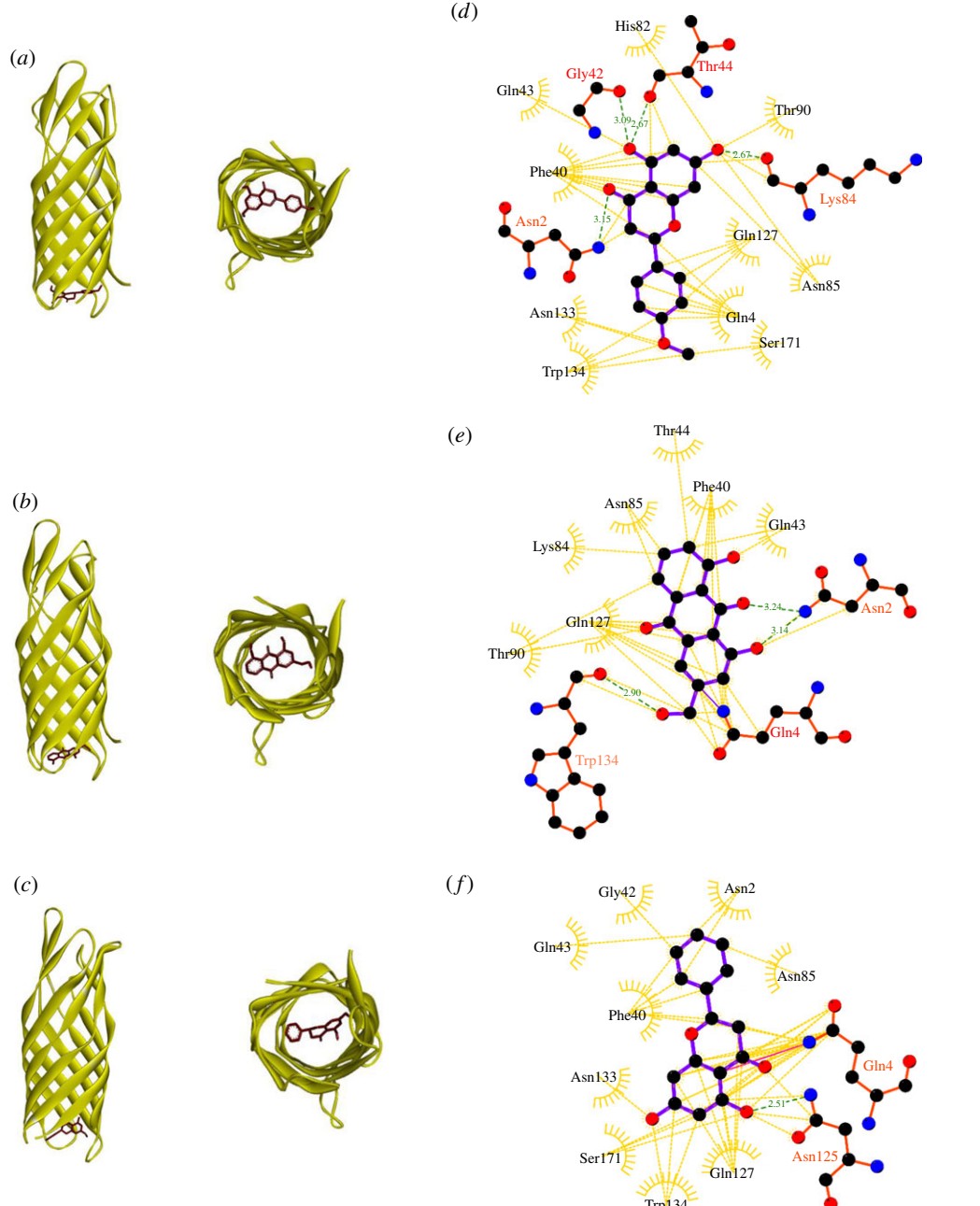

**Figure 6.** Three-dimensional representation of the compound-OmpW complexes; (*a*) represents isosakuranetin and OmpW complex, (*b*) aloe-emodin and OmpW complex, and (*c*) pinocembrin and OmpW complex; (*d*), (*e*) and (*f*) represent two-dimensional interaction diagram of isosakuranetin, aloe-emodin and pinocembrin in the binding pockets of the target, respectively. Ligands, H-bond interacting residues and hydrophobic interacting residues are shown in purple, red and golden, respectively. Green bonds represent hydrogen bonds and golden bonds represent hydrophobic interactions.

in the inner side of the OmpW β-barrel structure and interact with its residues. This can block the channel provided by this porin which consequently prevents nutrients from entering the cell.

Top three binding cavities in the energy minimized homology models of OmpA and OmpW were predicted by the METAPOCKET target prediction tool (figure 4). Results showed that from eight methods used by the server, only Q-SITEFINDER and POCASA algorithms failed to identify binding sites within the top predictions for both proteins. The cavity ID 03, predicted as one of OmpA binding pockets, was selected as a potential target, based on AOA-2 interaction with external loops of β-barrel structure [24]. In the case of OmpW, the cavity ID 02, containing the residues of the inner side of the β-barrel structure, was selected based on its potential interaction with D6 [32].

**Table 3.** The physico-chemical properties of the best selected compounds with mutual inhibitory effect on both OmpA and OmpW.

| compound name | $n_{Rot}$[a] | HBD[b] | HBA[c] | TPSA[d] (Å2) | logP[e] | MR[f] | Lipinski violation | CMC-like rule | lead-like rule |
|---|---|---|---|---|---|---|---|---|---|
| isosakuranetin | 2 | 2 | 5 | 120.92 | 2.81 | 76.04 | 0 | qualified | suitable |
| aloe-emodin | 1 | 3 | 5 | 113.28 | 1.36 | 69.92 | 0 | qualified | suitable |
| pinocembrin | 1 | 2 | 4 | 109.44 | 2.8 | 69.55 | 0 | qualified | suitable |

[a]number of rotatable bonds.
[b]hydrogen bond donor.
[c]hydrogen bond acceptor.
[d]topological polar surface area.
[e]lipophilicity.
[f]molar refractivity.

The molecular docking results showed that isosakuranetin, aloe-emodin and pinocembrin, with the lowest binding energy (−7.8, −7.7 and 7.5, respectively), are top three ligands to interact with the selected binding pocket of OmpA (table 1). Although the affinity of these compounds for OmpA is lower than AOA-2, but in this regard, they can be served as stronger interacting ligands than episterol (an *A. baumannii* growth inhibitor with the ability of binding to OmpA [9]). Both isosakuranetin and pinocembrin interact with residues Gln14, Asp15, Leu27, Thr28, Ser30, Glu158 and Trp160 (figure 5). Both compounds can interact with Asp15 by forming hydrogen bonds. In this regard, aloe-emodin is slightly different from isosakuranetin and pinocembrin. Aloe-emodin has no interaction with Ser30 and forms hydrogen bonds with Asp15 and Gln14.

Compared to D6, all the above-mentioned compounds possess a higher affinity for OmpW (table 2). Isosakuranetin interacts with the highest number of residues in the selected binding pocket of OmpW. Four out of 14 residues in OmpW (Asn2, Gln42, Thr44 and Lys 84) form hydrogen bonds with isosakuranetin. Aloe-emodin and pinocembrin interact with 10 and 11 residues of OmpW, respectively. Aloe-emodin forms hydrogen bonds with three residues (Asn2, Gln4 and Trp134), while pinocembrin only possesses one H-bond connected to Gln4 (figure 6).

## 3.5. Pharmacokinetic properties of selected hit phytochemicals

Physico-chemical, ADMET and safety endpoints of the selected hit phytochemicals were predicted (tables 2, 3 and 4). Results showed that all of these compounds satisfy the key parameters of a drug molecule via passing in Lipinski's RO5 (table 3). Analysing their ADMET profile, we observed that all of the hit inhibitors possess good GI (74.179–92.417%). Among them, the compounds isosakuranetin and pinocembrin with log Papp values of 1.1 and 1.52, respectively, showed high Caco-2 permeability. $LD_{50}$ values of the compounds were in the range of 1.586–2.329 mol kg$^{-1}$. None of the compounds was observed as OCT2 substrate. None of them acts as CYP2D6 substrates/inhibitors (table 4). Additionally, the selected hits were predicted as compounds with no inhibitory effect on hERGI. They also showed no hepatotoxicity and skin sensitization activity. The high value of VDss predicted for aloe-emodin ($\log_{L/kg} = 0.671 > 0.45$), indicates that most amount of the drug can be contained in tissues (rather than in plasma). Since only the unbound (free) drug can diffuse between plasma and tissues, the ability of the selected compounds to interact with pharmacological target proteins such as channels was also examined. The values (Fu) were in the range of 0.022–0.226 (table 5).

Apart from efficacy and toxicity, poor PK and bioavailability are among important factors of drug development failures. Among these, GI and brain access are crucial PK features, which need to be estimated in drug discovery processes. Permeation from the BBB, as a physical and biochemical shield protecting the brain, is the basic of the distribution of central-acting molecules. Moreover, regarding the preference of oral bioavailability of a drug in drug administration, human intestinal absorption (HIA) is an important roadblock in drug research. The accurate predictive model of Brain Or IntestinaL EstimateD permeation method (BOILED-Egg) was used to predict these two important behaviours for selected phytocompounds (figure 7). The model calculates the lipophilicity and polarity of small molecules. It allows for the evaluation of HIA as a function of the position of the small molecules in the WLOGP-versus-TPSA referential. The white region of the 'BOILED-Egg' is for a high probability of

**Table 4.** ADMET properties of the best selected compounds with mutual inhibitory effect on both OmpA and OmpW.

| compound name | ID | absorption | | distribution | | metabolism | excretion | toxicity | |
| | | GI (%) | water solubility (log mol l$^{-1}$) | log Papp | log BB | log PS | CYP2D6 Subs./Inh. | OCT2 Subs | AMES toxicity | LD$_{50}$ (mol kg$^{-1}$) |
|---|---|---|---|---|---|---|---|---|---|---|
| isosakuranetin | 160 481 | 92.37 | −3.092 | 1.1 | 0.194 | −2.17 | no/no | no | no | 1.843 |
| aloe-emodin | 10 207 | 74.179 | −3.104 | −0.233 | −0.729 | −2.466 | no/no | no | yes | 2.329 |
| pinocembrin | 68 071 | 92.417 | −3.538 | 1.152 | 0.42 | −2.047 | no/no | no | no | 1.586 |

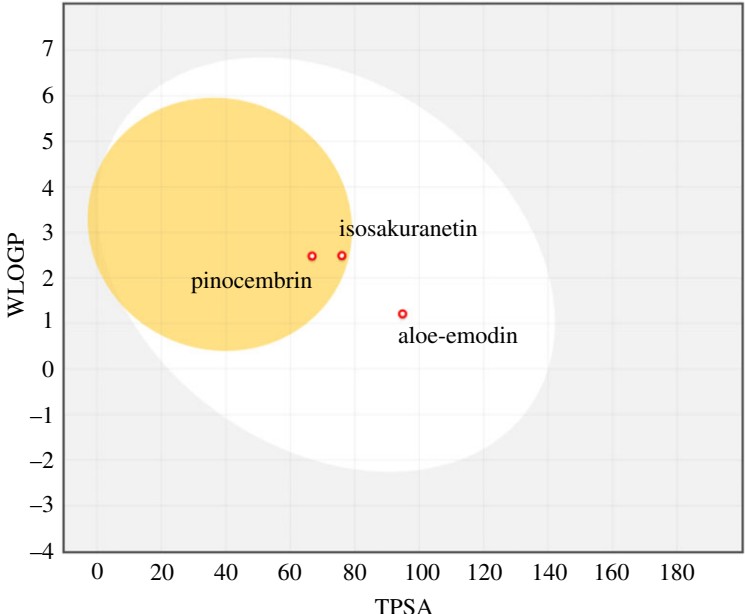

**Figure 7.** The BOILED-Egg model of the selected best potential inhibitors of OmpA and OmpW. The compounds located in the yellow region were predicted to be passively penetrated from BBB. The compound in the white region is predicted to be passively absorbed by gastrointestinal tract. The molecules predicted to be effluated from the CNS by p-glycoproteins are shown by blue dots. Those which are not effluated from the CNS by P-glycoproteins are shown by red dots.

**Table 5.** The computed safety endpoints of the best selected compounds with mutual inhibitory effect on both OmpA and OmpW.

| compound name | total clearance | hERGI inhibitor | oral rat chronic toxicity (LOAEL) | hepatotoxicity | skin sensitization | VDss[a] | $F_u$[b] |
|---|---|---|---|---|---|---|---|
| isosakuranetin | 0.111 | no | 2.148 | no | no | 0.217 | 0.077 |
| aloe-emodin | 0.008 | no | 1.878 | no | no | 0.671 | 0.226 |
| pinocembrin | 0.122 | no | 2.059 | no | no | −0.386 | 0.022 |

[a]steady-state volume of distribution (VDss).
[b]fraction unbound ($F_u$).

passive absorption of a compound by the gastrointestinal tract, and the yellow region (yolk) is for a high probability of brain penetration. Yolk and white areas are not mutually exclusive. In addition, the points are coloured in blue if a compound is predicted as actively effluxed by P-glycoprotein (PGP+) and in red if a compound is predicted as non-substrate of P-gp (PGP−). Our results showed that among the selected compounds, isosakuranetin and pinocembrin were predicted to permeate through BBB by passive diffusion. According to the prediction, aloe-emodin can be passively absorbed by the gastrointestinal tract. The assessment of P-gp efflux for the selected compounds showed all of the selected compounds are not substrates of P-gp. Indeed, P-glycoprotein is a multi-drug transporter, which is apically expressed in the gastrointestinal tract, liver, kidney and brain endothelium [52]. Consequently, P-glycoprotein plays an important role in the oral bioavailability, CNS distribution biliary and renal elimination of drugs which are substrates of this transporter.

HIA and BBB are dependent on the water solubility and lipophilicity of the drug. Two topological methods of the ESOL [53] and Ali [54] are included in SwissADME to predict water solubility. SwissADME third predictor for solubility was developed by SILICOS-IT. All predicted values are the decimal logarithm of the molar solubility in water (log S) (table 5). Three predictors classified aloe-emodin and pinocembrin as soluble molecules. Isosakuranetin is predicted to be soluble adopted by ESOL predictor, but moderately soluble by Ali and SILICOS-IT predictors. According to log p, isosakuranetin (log p = 2.81) and pinocembrin (log p = 2.8) are more lipophilic than aloe-emodin (log p = 1.36) (table 6).

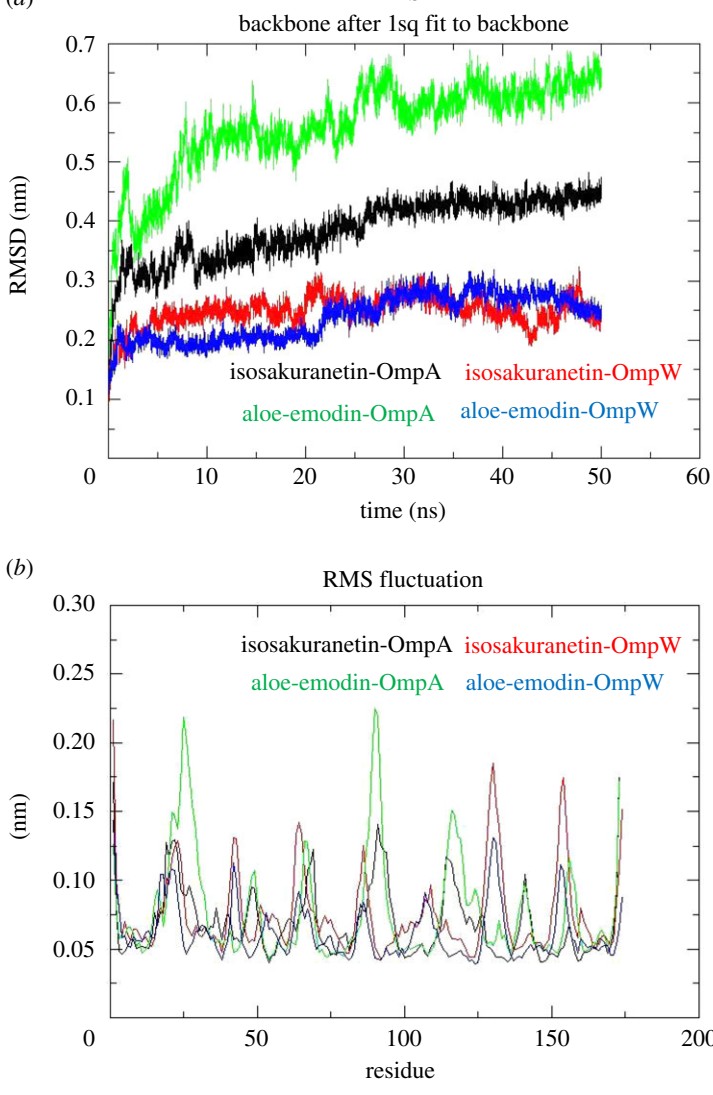

**Figure 8.** Detection of the stability of the protein–ligand complexes. (*a*) RMSD of the protein–ligand complexes. (*b*) RMSF plots of the protein–ligand complexes. Isosakuranetin bounded to OmpA (black) and bounded to OmpW (red). Aloe-emodin bounded to OmpA (green) and bounded to OmpW (blue).

**Table 6.** The water solubility evaluation of the best selected compounds with mutual inhibitory effect on both OmpA and OmpW.

| molecule | ESOL LogS | Ali LogS | SILICOS-IT LogS | ESOL class | Ali class | SILICOS-IT class |
|---|---|---|---|---|---|---|
| isosakuranethin | −3.7 | −4.1 | −4.12 | soluble | moderately soluble | moderately soluble |
| aloe-emodin | −3.04 | −3.43 | −3.92 | soluble | soluble | soluble |
| pinocembrin | −3.64 | −3.94 | −4.00 | soluble | soluble | soluble |

## 3.6 Molecular dynamics simulation analysis

Given the results, it seems isosakuranetin and aloe-emodin can be served as suitable ligands for both OmpA and OmpW to inhibit their targeted functions. Therefore, both of the compounds were selected to evaluate their molecular docking results using MD simulation (figure 8). Backbone deviation of complexes was plotted as a function of time. The results showed stability of complexes. The aloe-emodin-OmpA

R. Soc. Open Sci. **8**: 201652

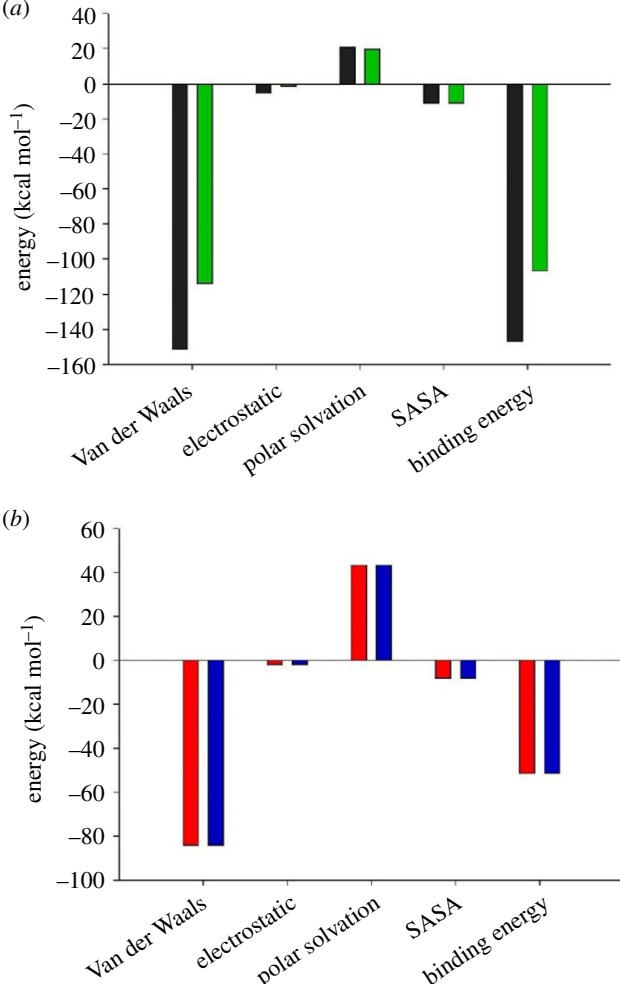

**Figure 9.** (*a*) Binding energies (including its components) of isosakuranetin-OmpA and aloe-emodin-OmpA (black and green bars, respectively). (*b*) Binding energies (including its components) of isosakuranetin-OmpW and aloe-emodin-OmpW (red and blue bars, respectively).

complex showed a RSMD value ranging from 0.1 to 0.65 nm during simulation. RMSD value for the isosakuranetin-OmpA complex was in the range of 0.1 to 0.65 nm. Both the isosakuranetin- and aloe-emodin-OmpW complex demonstrated a similar value, ranging from 0.1 to 0.4 nm. The flexibility of backbone was evaluated using root mean square fluctuation (RMSF) plot. The higher RMSF value means the higher flexibility, whereas the lower value represents the lower mobility during simulation. In general, β-strand residues showed less fluctuations than loops. Compared to the aloe-emodin-OmpA complex, the isosakuranetin-OmpA complex demonstrated lower fluctuations. In regard to the OmpW-ligand complexes, the isosakuranetin-OmpW complex showed slightly higher fluctuations than the aloe-emodin-OmpW complex (figure 9).

We performed MM/PBSA analysis to evaluate free-binding energy of the ligand–protein complexes. Compared to the aloe-emodin, isosakuranetin interacts with OmpA with larger Van der Waals and electrostatic energy value. The binding energies for isosakuranetin and aloe-emodin, in the form of OmpA-ligand complex, were −147.097 and −106.871 kcal mol$^{-1}$, respectively. According to OmpW-ligand complex, both isosakuranetin and aloe-emodin form complexes with the same binding energy (−51.49 kcal mol$^{-1}$). Overall, although the results show that both compounds are suitable for the purpose of our study, isosakuranetin appears to establish more stable interactions with both OmpA and OmpW.

## 4. Conclusion

The global AMR problem raises the necessity of novel approaches for the development of new drugs against emerging superbugs. A major contributor to this resistance is the outer membrane of

Gram-negative bacteria with OMPs as promising drug targets due to their extensive functions. Therefore, any molecule which has the potential to disrupt the normal activity of these porins can help the host defence system overcoming the pathogens. Our results suggest three out of 384 phytocompounds (isosakuranetin, aloe-emodin and pinocembrin) and can control *A. baumannii* infection by targeting the function of two proteins, OmpA, based on its important roles in biofilm formation and bacterial adhesion to host cells, and OmpW, based on its important role in the transport of essential nutrients. Also, these compounds showed promising PK features as potential drugs with possible anti-*A. baumannii* activities. Among them, isosakuranetin was introduced as a top inhibitor for the targeted functions of OmpA and OmpW. Therefore, this biomolecule can be further characterized experimentally to corroborate its inhibitory activity. In addition, the skeletons of these active compounds could be adopted as starting potential scaffolds for the design of future anti-*A. baumannii* drugs.

Data accessibility. The data supporting the results in this article can be accessed at the Dryad Digital Repository: https://doi.org/10.5061/dryad.k6djh9w6f.

The data are provided in the electronic supplementary material [55].

Authors' contributions. S.S. conceived and designed the experiments; S.S. and P.M. performed the experiments; S.S., P.M. and K.A.N. analysed the data and coordinated the study; S.S. and P.M. drafted the manuscript. S.S. and K.A.N. revised the manuscript. All authors gave final approval for publication and agree to be held accountable for the work performed therein.

Competing interests. The authors declare no competing interests.

Funding. The authors received no financial support for this study.

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
