## [Peer Review File · Royal Society Open Science]

Review History

RSOS-201652.R0 (Original submission)

Review form: Reviewer 1

Is the manuscript scientifically sound in its present form?

No

Are the interpretations and conclusions justified by the results?

No

Is the language acceptable?

No

Do you have any ethical concerns with this paper?

No

Have you any concerns about statistical analyses in this paper?

No

Recommendation?

Reject

Comments to the Author(s)

The manuscript titled "Screening of anti-Acinetobacter baumannii plant-based compounds, based on potential inhibition of OmpA and OmpW functions" is important and it deals with the identification of potential plant-based compounds towards the OmpA and OmpW of Acinetobacter baumannii. As MDR Acinetobacter baumannii is one of the potential issues world wide and WHO reported the organism is priority-1 pathogen, the present study has significant application.

The major issue is that objectives are not clearly mentioned and the background data is not adequately highlighted. The authors have missed several papers that was published recently with the same theme. All these papers need to discuss and compare their finding with the existing literature and state the novelty of eth present work

Crit Rev Microbiol. 2019 45(3):315-333.doi: 10.1080/1040841X.2019.1600472.

Infect Genet Evol. 2020 82:104314.doi: 10.1016/j.meegid.2020.104314. Epub 2020 Apr 5.

J Biomol Struct Dyn. 2019 Mar;37(5):1146-1169. doi: 10.1080/07391102.2018.1451387

Line 110-National Center for Biotechnology 110 Information (NCBI) protein database. It is not NCBI database, It is GenBank database, which is hosted at NCBI

The 3D modeling of the OmpA and OmpW are not clearly discussed. What is the target, templates, selection of best templates, critical parameters considered? structure annotated pair wise alignments, loop modeling, side chain optimization if required etc. are not properly discussed. Similarly, the refinement and validation are performed by limited tools, and the interpretation is not ideally discussed.

The virtual screening needs to be strengthened by utilizing several tools for drug likeliness predictions, only Lipinski rule of five is not satisfactory. When you use Lipinski rule, most of the molecules will qualify that, and the further screening is become tough. Thus, it is advised to use several other filters and make a very adequate and accurate screening.

In molecular docking, the preparation of ligands and receptors are not highlighted? How the grid box for the binding cavity is prepared?

There are several theoretical aspects are highlighted in the methodology throughout the manuscript. Remove the theoretical statements from the methodology and only highlight the references whatever it is required.

There are no comparative controls shown for the docking studies. Without the control the prediction showed less accuracy

There are no MD simulation and energy calculations by MMPBSA/MMGBSA are highlighted. Without the MD simulation, the prediction is not satisfactory and unacceptable.

Review form: Reviewer 2**Is the manuscript scientifically sound in its present form?**

Yes

Are the interpretations and conclusions justified by the results?

Yes

Is the language acceptable?

Yes

Do you have any ethical concerns with this paper?

No

Have you any concerns about statistical analyses in this paper?

Yes

Recommendation?

Accept with minor revision (please list in comments)

Comments to the Author(s)

The authors have done an extensive computational work in the manuscript.

The results have been well clarified.

The authors only need to check the paper for English language errors and grammatical mistakes.

I recommend this paper for acceptance in its current form after english language polishing.

Decision letter (RSOS-201652.R0)

Dear Mr Shahryari:

Title: Screening of anti-Acinetobacter baumannii plant-based compounds, based on potential inhibition of OmpA and OmpW functions

Manuscript ID: RSOS-201652

The editor assigned to your manuscript has now received comments from reviewers. We would like you to revise your paper in accordance with the referee and Subject Editor suggestions which can be found below (not including confidential reports to the Editor). Please note this decision does not guarantee eventual acceptance.

Please submit your revised paper before 02-May-2021. Please note that the revision deadline will expire at 00.00am on this date. If we do not hear from you within this time then it will be assumed that the paper has been withdrawn. In exceptional circumstances, extensions may be possible if agreed with the Editorial Office in advance. We do not allow multiple rounds of revision so we urge you to make every effort to fully address all of the comments at this stage. If deemed necessary by the Editors, your manuscript will be sent back to one or more of the original reviewers for assessment. If the original reviewers are not available we may invite new reviewers.

When submitting your revised manuscript, you must respond to the comments made by the referees and upload a file "Response to Referees" in "Section 6 - File Upload". Please use this to document how you have responded to the comments, and the adjustments you have made. In

order to expedite the processing of the revised manuscript, please be as specific as possible in your response.

On behalf of the Subject Editor Professor Anthony Stace and the Associate Editor Dr Debashree Ghosh.

RSC Associate Editor:

Comments to the Author:

The reviewers have suggested substantial modifications and have concerns which need to be clarified in a point wise manner.

RSC Subject Editor:

Comments to the Author:

(There are no comments.)

Reviewers' Comments to Author:

Reviewer: 1

Comments to the Author(s)

The manuscript titled "Screening of anti-Acinetobacter baumannii plant-based compounds, based on potential inhibition of OmpA and OmpW functions" is important and it deals with the identification of potential plant-based compounds towards the OmpA and OmpW of Acinetobacter baumannii. As MDR Acinetobacter baumannii is one of the potential issues world wide and WHO reported the organism is priority-1 pathogen, the present study has significant application.

The major issue is that objectives are not clearly mentioned and the background data is not adequately highlighted. The authors have missed several papers that was published recently with the same theme. All these papers need to discuss and compare their finding with the existing literature and state the novelty of eth present work

Crit Rev Microbiol. 2019 45(3):315-333.doi: 10.1080/1040841X.2019.1600472.

Infect Genet Evol. 2020 82:104314.doi: 10.1016/j.meegid.2020.104314. Epub 2020 Apr 5.

J Biomol Struct Dyn. 2019 Mar;37(5):1146-1169. doi: 10.1080/07391102.2018.1451387

Line 110-National Center for Biotechnology 110 Information (NCBI) protein database. It is not NCBI database, It is GenBank database, which is hosted at NCBI

The 3D modeling of the OmpA and OmpW are not clearly discussed. What is the target, templates, selection of best templates, critical parameters considered? structure annotated pairwise alignments, loop modeling, side chain optimization if required etc. are not properly discussed. Similarly, the refinement and validation are performed by limited tools, and the interpretation is not ideally discussed.

The virtual screening needs to be strengthened by utilizing several tools for drug likeliness predictions, only Lipinski rule of five is not satisfactory. When you use Lipinski rule, most of the molecules will qualify that, and the further screening is become tough. Thus, it is advised to use several other filters and make a very adequate and accurate screening.

In molecular docking, the preparation of ligands and receptors are not highlighted? How the grid box for the binding cavity is prepared?

There are several theoretical aspects are highlighted in the methodology throughout the manuscript. Remove the theoretical statements from the methodology and only highlight the references whatever it is required.

There are no comparative controls shown for the docking studies. Without the control the prediction showed less accuracy

There are no MD simulation and energy calculations by MMPBSA/MMGBSA are highlighted. Without the MD simulation, the prediction is not satisfactory and unacceptable.

Reviewer: 2

Comments to the Author(s)

The authors have done an extensive computational work in the manuscript.

The results have been well clarified.

The authors only need to check the paper for English language errors and grammatical mistakes.

I recommend this paper for acceptance in its current form after english language polishing.

Author's Response to Decision Letter for (RSOS-201652.R0)

See Appendix A.

RSOS-201652.R1 (Revision)

Review form: Reviewer 1

Is the manuscript scientifically sound in its present form?

Yes

Are the interpretations and conclusions justified by the results?

Yes

Is the language acceptable?

Yes

Do you have any ethical concerns with this paper?

No

Have you any concerns about statistical analyses in this paper?

No

Recommendation?

Accept as is

Comments to the Author(s)

The author shave satisfactorily revised the manuscript, whatever the issues I have raised in my previous reviews, the authors have addressed. The scientific quality of the manuscript has been improved. Thus, the present form of the paper can be considered for publication.

Review form: Reviewer 2

Is the manuscript scientifically sound in its present form?

Yes

Are the interpretations and conclusions justified by the results?

Yes

Is the language acceptable?

Yes

Do you have any ethical concerns with this paper?

No

Have you any concerns about statistical analyses in this paper?

No

Recommendation?

Accept as is

Comments to the Author(s)

The authors have satisfactorily addressed all the comments. the article is acceptable now.

Decision letter (RSOS-201652.R1)

Dear Mr Shahryari:

Title: Screening of anti-Acinetobacter baumannii phytochemicals, based on the potential inhibitory effect on OmpA and OmpW functions

Manuscript ID: RSOS-201652.R1

It is a pleasure to accept your manuscript in its current form for publication in Royal Society Open Science. The chemistry content of Royal Society Open Science is published in collaboration with the Royal Society of Chemistry.

On behalf of the Subject Editor Professor Anthony Stace and the Associate Editor Dr Debashree Ghosh.

RSC Associate Editor:
Comments to the Author:
The manuscript is acceptable as the authors have made the modifications and answered the queries by the referees.

RSC Subject Editor:
Comments to the Author:
(There are no comments.)

Reviewer(s)' Comments to Author:
Reviewer: 1

Comments to the Author(s)
The author shave satisfactorily revised the manuscript, whatever the issues I have raised in my previous reviews, the authors have addressed. The scientific quality of the manuscript has been improved. Thus, the present form of the paper can be considered for publication.

Reviewer: 2
Comments to the Author(s)
The authors have satisfactorily addressed all the comments. the article is acceptable now.

Appendix A

23-May-2021

Dear respectable editor,

We would like to thank you and reviewers for commenting on our manuscript “Screening of anti-*Acinetobacter baumannii* phytochemicals, based on the potential inhibitory effect on OmpA and OmpW functions” (RSOS-201652). All the comments and suggestions have certainly improved the quality of our manuscript. We have tried to consider all the reviewer’s comments point by point as far as possible and hope that you will accept our paper for publication in journal of Royal Society Open Science. The following letter gives our response to the referee’s comments as requested in your letter of 09 April 2021.

Thank you again for your continuous support,

Highest regards

Shahab Shahryari, Ph.D

Research associate

Division of Industrial & Environmental Biotechnology, National Institute of Genetic Engineering and Biotechnology (NIGEB), P.O.BOX 14155-6343, Tehran-Iran.

Response to Reviewer 1:

The manuscript titled “Screening of anti-Acinetobacter baumannii plant-based compounds, based on potential inhibition of OmpA and OmpW functions” is important and it deals with the identification of potential plant-based compounds towards the OmpA and OmpW of Acinetobacter baumannii. As MDR Acinetobacter baumannii is one of the potential issues world wide and WHO reported the organism is priority-1 pathogen, the present study has significant application.

1) The major issue is that objectives are not clearly mentioned and the background data is not adequately highlighted. The authors have missed several papers that was published recently with the same theme. All these papers need to discuss and compare their finding with the existing literature and state the novelty of the present work

Crit Rev Microbiol. 2019 45(3):315-333.doi: 10.1080/1040841X.2019.1600472.

Infect Genet Evol. 2020 82:104314.doi: 10.1016/j.meegid.2020.104314. Epub 2020 Apr 5.

J Biomol Struct Dyn. 2019 Mar;37(5):1146-1169. doi: 10.1080/07391102.2018.1451387

The novelty of this study is based on a new strategy to mutually disrupt the activity of OmpA and OmpW. In this regard, the adhesion property of OmpA and the function of OmpW to uptake the essential nutrients were targeted in this study. This study also suggests new plant based compounds with ability to cripple *A. baumannii* with multi-target drug properties.

As the study focuses on OmpA and OmpW, epiestriol was selected for comparison with our results, according to its potential inhibitory effect on OmpA (as discussed by one of the papers suggested by reviewer doi: 10.1016/j.meegid.2020.104314. Epub 2020 Apr 5). However, the other suggested

papers could help us to improve the introduction section. To make the aim and the novelty of the study more clear, the summary, introduction, and conclusion was re-written in a more transparent form.

2) Line 110-National Center for Biotechnology Information (NCBI) protein database. It is not NCBI database, It is GenBank database, which is hosted at NCBI

Thank you very much. It was considered. Please refer to line 115 in page 5 of edited version of manuscript.

3) The 3D modeling of the OmpA and OmpW are not clearly discussed. What is the target, templates, selection of best templates, critical parameters considered? structure annotated pair wise alignments, loop modeling, side chain optimization if required etc. are not properly discussed. Similarly, the refinement and validation are performed by limited tools, and the interpretation is not ideally discussed.

After 3D structure prediction of proteins, the structures submitted to GalaxyRefine web server for structure refinement. Then, the total energy of structure was minimized using YASARA energy minimization server. The tertiary structure was validated with RAMPAGE, ProSA-web and ERRAT and all the necessary outputs which can show the quality of our structure have been discussed in a separate section (please refer to “3.2. tertiary structure validation”).

This is a routine process which is used in most articles. On the other hand, there is no theoretical consensus on the use of several refinement methods to obtain a better final structure (Please refer to <https://doi.org/10.3390/ijms20092301>). Further refinement does not mean better 3D model. It can even lead to a degradation in the quality of structures. Therefore, as a routine process which is used in most studies, we performed the steps mentioned above. Then after ensuring the quality of

the models (using RAMPAGE, ProSA-web and ERRAT), we used the models for further study.

4) The virtual screening needs to be strengthened by utilizing several tools for drug likeliness predictions, only Lipinski rule of five is not satisfactory. When you use Lipinski rule, most of the molecules will qualify that, and the further screening is become tough. Thus, it is advised to use several other filters and make a very adequate and accurate screening.

Thank you very much for this comment. In order to screen the compounds with high accuracy, in the new version of manuscript, in addition to SwissADME and pkCSM servers, PreADMET server was also considered (as suggested in [doi: 10.1016/j.meegid.2020.104314](https://doi.org/10.1016/j.meegid.2020.104314). Epub 2020 Apr 5). Therefore the compounds were screened based on more drug likeliness filters. Please refer to lines 172 and 173, page 7.

5) In molecular docking, the preparation of ligands and receptors are not highlighted? How the grid box for the binding cavity is prepared?

It was added in the edited version of manuscript. Please refer to lines 174 to 187, pages 7 & 8, section: 2.9. Molecular docking analysis).

6) There are several theoretical aspects are highlighted in the methodology throughout the manuscript. Remove the theoretical statements from the methodology and only highlight the references whatever it is required.

In order to this comment, the section “2. Data and methodology” (lines 109-213, pages 5-9) was re-written in the new version of manuscript.

7) There are no comparative controls shown for the docking studies. Without the control the prediction showed less accuracy

In this regard, we compared our results with AOA-2 and D6 to further validate or data.

For details please refer to 3.4. Protein-Ligand Interactions and Mechanisms of Binding

(lines 318-332, pages 13 & 14).

8) There are no MD simulation and energy calculations by MMPBSA/MMGBSA are highlighted. Without the MD simulation, the prediction is not satisfactory and unacceptable.

Thank you for this suggestion. A MD simulation analysis section was added in the new version of manuscript. Please refer to lines 398-420 in pages 16 and 17.

Reviewer: 2

Comments to the Author(s)

The authors have done an extensive computational work in the manuscript.

The results have been well clarified.

The authors only need to check the paper for English language errors and grammatical mistakes.

I recommend this paper for acceptance in its current form after english language polishing.

I would like to thank the esteemed reviewer for his/her encouragement words. The English language polishing was performed all around the text and figure captions.